# Clinical Outcomes of Stereotactic Ablative Radiotherapy for All Stages of Non-Small Cell Lung Cancer; Definitive versus Consolidative

**DOI:** 10.3390/medicina58091304

**Published:** 2022-09-18

**Authors:** Hakyoung Kim, Sun Myung Kim, Dae Sik Yang, Kyung Hwa Lee, Young Bum Kim

**Affiliations:** Departments of Radiation Oncology, Korea University Guro Hospital, Korea University College of Medicine, Seoul 08308, Korea

**Keywords:** non-small cell lung cancer, stereotactic ablative radiotherapy, survival, local control

## Abstract

*Background and Objectives:* Stereotactic ablative radiotherapy (SABR) is not confined to early stage non-small cell lung cancer (NSCLC) and has a potential role in stage IV disease. We aimed to evaluate the effect of SABR on local control rates and survival outcomes in patients with all stages of NSCLC according to the treatment aim. *Materials and Methods:* We retrospectively reviewed the medical records of 88 patients with NSCLC who received SABR at the Korea University Guro Hospital between January 2015 and March 2021. Among these, 64 patients with stage I–II NSCLC ineligible for surgery were treated with a definitive aim. Twenty-four patients with stage IV limited metastatic NSCLC showing a favorable response to prior systemic therapy were treated with a consolidative aim. *Results:* The median follow-up time was 34 (range: 5–88) months. Thirty-one patients developed recurrence (35.2%), with distant metastasis being the most common (25/31, 80.6%). In-field local recurrence occurred in four patients (4/88 patients, 4.5%). For patients treated with definitive SABR, the 3-year overall survival (OS) and disease-free survival (DFS) rates were 91.8% and 58.6%, respectively. In patients treated with consolidative SABR, the 3-year OS and DFS rates were 86.7% and 53.8%, respectively. With respect to treatment-related pulmonary toxicity, grade 3 radiation pneumonitis incidence requiring hospitalization was 2.3% (2/88). *Conclusions:* Definitive SABR is appropriate for medically inoperable or high surgical risk patients with early stage NSCLC with acceptable treatment-related toxicities. Consolidative SABR improves local control rates and helps achieve long-term survival in patients with limited metastatic NSCLC.

## 1. Introduction

Stereotactic ablative radiotherapy (SABR) has shown favorable outcomes with a reduced risk of local recurrence compared with wedge resection for patients with stage I non-small cell lung cancer (NSCLC) who are ineligible for anatomic lobectomy [1,2,3,4]. In addition, two randomized phase III trials, STARS and ROSEL [5], showed promising results for SABR with similar survival outcomes and loco-regional control rates compared with lobectomy with mediastinal lymph node dissection in patients with operable stage I NSCLC. The current National Comprehensive Cancer Network (NCCN) guidelines also recommend SABR as an appropriate option for patients who are medically inoperable or at high surgical risk. However, objective criteria for the feasibility of surgical resection remain unclear. A considerable number of patients who are unable to tolerate lobectomy due to various factors such as old age, poor performance status, impaired lung function, and comorbidities are treated with wedge resection or segmentectomy according to the institutional preference without sufficient explanation regarding alternative options.

Recently, SABR has not been confined to patients with an early stage disease who are medically inoperable or high-risk surgical candidates. It has a potential role in all stages of NSCLC, especially in patients with stage IV disease who have shown a favorable response to prior systemic therapy. Remarkable advances in targeted agents and immuno-oncologic agents have provoked great interest in consolidative local therapy for various types of cancers [6,7,8,9,10,11,12,13,14]. In particular, aggressive local therapy using surgical resection or high-dose radiotherapy has been proven to improve local control rates and to provide significant survival benefits in patients with limited metastatic NSCLC [15,16,17].

The purpose of this study was to evaluate the effect of SABR on local control rates and survival outcomes in patients with all stages of NSCLC according to the treatment aim.

## 2. Materials and Methods

### 2.1. Patients

After obtaining institutional review board approval (#2022GR0196), we retrospectively reviewed the medical records of 88 patients with NSCLC who received SABR, with a total dose of 60 Gy in four fractions, at the Korea University Guro Hospital between January 2015 and March 2021. Among these, 64 patients with stage I–II NSCLC who were not eligible for surgery for various reasons, such as old age, poor performance status, impaired lung function, and comorbidities, were treated with a definitive aim. The remaining 24 patients with stage IV, limited metastatic NSCLC who had shown a partial response, stable disease, or oligo-progressive disease at the thoracic site in their tumor response evaluation to prior systemic therapy were treated with a consolidative aim. Limited metastasis was undefined, but the clinical trials included three to five metastases [16,17,18]. Specifically, we included patients with one to three limited metastatic sites at the time of diagnosis. All tumors were staged according to the 8th edition of the American Joint Committee on Cancer Tumor Staging Criteria.

### 2.2. Treatment Scheme

All of the patients underwent SABR for treatment planning. Specifically, SABR was applied to small (≤4 cm) and peripherally located tumors, with a total dose of 60 Gy in four fractions.

Regarding target delineation, the gross tumor volume (GTV) was delineated under the lung window setting. The internal target volume (ITV) was delineated following four-dimensional CT, with special regard to the patient’s respiratory motion. The clinical target volume (CTV) was generated with a 5 mm expansion of the GTV-ITV in all directions and then modified according to the adjacent normal anatomic structures. The planning target volume was generated with a 5 mm expansion of the CTV. The prescription guideline was to deliver at least 97% of the prescribed dose to 95% of the CTV.

### 2.3. Surveillance

Chest computed tomography scans were performed every 3 months for 2 years after SABR to detect disease progression during follow-up. The Revised Response Evaluation Criteria in the Solid Tumors guidelines (Version 1.1) were used to evaluate tumor response. Treatment-related pulmonary complications, excluding infection cases, were evaluated using the Common Terminology Criteria for Adverse Events version 4.03.

### 2.4. Statistical Analysis

The data were reported as numbers for categorical variables and medians for continuous variables. Overall survival (OS) was defined as the time from the start of radiotherapy (in the definitive group) or the start of systemic therapy (in the consolidative group) until the date of death from any cause or to the latest documented follow-up. Disease-free survival (DFS) was defined as the time from the start of radiotherapy (in the definitive group) or the start of systemic therapy (in the consolidative group) until the date of the first documented disease progression or the latest documented follow-up. The 3-year OS and DFS rates were calculated using the Kaplan–Meier method. Statistical analyses were performed using IBM SPSS Statistics for Windows (version 24.0; IBM Corporation, Armonk, NY, USA).

## 3. Results

### 3.1. Baseline Characteristics

The clinical characteristics of the patients according to the treatment aim are summarized in Table 1. In the definitive group, the median age of the study population was 80 years (range: 61–91 years). More than half of the patients were male (59.4%), were either current or ex-smokers (50.0%), and had adenocarcinoma of the histologic type (64.1%). In the consolidation group, the median age of the study population was 73 years (range: 53–89 years). Most patients were male (66.7%), were never smokers (62.5%), and had adenocarcinoma of the histologic type (66.7%). Among patients treated with a consolidative aim, the majority of patients had one metastatic site at the time of diagnosis (17/24 patients, 70.8%). Most (54.1%) patients were treated with platinum-doublet chemotherapy, and only six (25.0%) patients were treated with immuno-oncologic agents. The median time from the start of systemic treatment to SABR was 18 months (range: 3–39 months). Among the 24 patients, five (20.8%) showed partial response, 15 (62.5%) showed stable disease, and four (16.7%) showed oligo-progressive disease at the thoracic site only in their tumor response evaluation after systemic treatment.

### 3.2. Patterns of Failure

The median follow-up time from the start of initial treatment was 34 months (range: 5–88 months). Additionally, the median follow-up time from the start of SABR was 27 months (range: 5–74 months). Thirty-one (35.2%) patients developed recurrence. The patterns of failure (distant metastasis alone, locoregional recurrence alone, and both) according to the treatment aim are described in Figure 1. Distant metastasis was the most common recurrence (25/31, 80.6%). The most common site of distant metastasis was the lungs (13 patients), followed by the brain and bone (seven patients for both).

In terms of radiotherapy, in-field local recurrence occurred in four patients (4/88, 4.5%), and the clinical courses of patients who showed local recurrence after SABR are specifically described in Table 2. Three patients were ex-smokers and had a squamous cell histological type. Time to local recurrence varied, and two patients showed in-field local recurrence alone at the second visit after SABR. All patients were treated with immuno-oncologic agents or other chemotherapeutic agents and remained alive without further disease progression.

### 3.3. Survival Outcomes and Treatment-Related Complications

The 3-year OS and DFS rates of patients treated with definitive SABR were 91.8% and 58.6%, respectively. In patients treated with consolidative SABR, the 3-year OS and DFS rates were 86.7% and 53.8%, respectively. The median OS was not reached in both groups.

With respect to treatment-related pulmonary toxicity, the incidence of grade 2 radiation pneumonitis requiring steroid medication was 17.0% (15/88) and that of grade 3 radiation pneumonitis requiring hospitalization was 2.3% (2/88) (Table 3). Both patients were men and had a history of smoking. In addition, these patients showed impaired pulmonary function (Pre-SABR DLCO: 56% and 24%, respectively) and had underlying lung diseases such as chronic obstructive pulmonary disease (COPD) and idiopathic pulmonary fibrosis (IPF). Severe radiation pneumonitis occurred three months after the completion of SABR. Another patient who had underlying severe COPD with Global Initiative for Obstructive Lung Disease (GOLD) grade 3 (pre-SABR FEV_1_; 0.8 L, 34%) showed acute exacerbation of COPD at one month later after completion of SABR. All of these patients were alive without further disease progression. No grade 4 or 5 treatment-related complications were observed.

## 4. Discussion

Existing data on patient survival and local control rates indicate that SABR is a reasonable treatment option for patients with early stage NSCLC who are medically ineligible for anatomic lobectomy [1,2,3,4,5]. Although there are some differences between studies, SABR has shown a reduced risk of local recurrence compared with wedge resection (4.0% vs. 20.0%, *p* = 0.05) and similar cause-specific survival for early stage NSCLC patients who were medically inoperable [1]. Furthermore, two randomized phase 3 trials, STARS and ROSEL, have shown promising results for SABR with high OS (3-year OS rate, 95.0% vs. 79.0%, *p* = 0.037) and similar recurrence-free survival (3-year RFS rate, 86.0% vs. 80.0%, *p* = 0.537) compared with those with lobectomy in patients with operable, early stage NSCLC [5]. Despite several limitations of this study, the low survival after lobectomy may be due to pre-existing comorbidities worsened by the invasive procedure. In this regard, the current NCCN guidelines also recommend SABR as an appropriate option for medically inoperable patients with high surgical risk. Therefore, preoperative physiological risk assessments must be carefully performed [19]. Identifying patients at an increased risk of surgery and enabling informed consent by patients about appropriate alternative treatment options is crucial.

Our hospital is in an area with a low socioeconomic status, and the proportion of patients diagnosed at an advanced stage is higher than those in the early stages. In the early stage, wedge resection or segmentectomy is strongly recommended for patients who cannot tolerate lobectomy, with a lack of explanation about alternative options, such as SABR. Although the overall number of patients included in this study was small, it showed promising results that are concordant with the results of previous reports. In the definitive SABR group, the 3-year OS and DFS rates were 91.8% and 58.6%, respectively. In addition, in-field local recurrence occurred in only three patients (3/64, 4.69%).

With respect to consolidative SABR, the SABR for the comprehensive treatment of oligo-metastasis (SABR-COMET) study [11], a randomized phase II trial, demonstrated that the addition of SABR at all sites of the disease improved survival outcomes in 99 patients with breast, prostate, and lung cancers compared with the palliative care alone group. Specifically, the addition of SABR led to a remarkable improvement in OS (median OS, 28 months vs. 41 months), with a doubling of PFS (median PFS, 6 months vs. 12 months). Extended follow-up data also showed that the impact of SABR on OS (5-year OS rate, 17.7% vs. 42.3%, *p* = 0.006) and PFS (5-year PFS, not reached in arm 1 and 17.3% in arm 2, *p* = 0.001) was larger than that in the initial result and durable over time [20]. Another randomized phase II trial showed a significant progression-free survival (PFS) benefit of local consolidative therapy (LCT) for stage IV oligometastatic NSCLC [15]. Here, 49 patients with a favorable response to prior systemic therapy were randomly assigned to receive maintenance therapy or LCT at all active disease sites in a 1:1 ratio. The long-term results of this study confirmed that LCT significantly improved the OS (median OS, 17.0 months vs. 41.2 months) and PFS (median PFS, 4.4 months vs. 14.2 months) [16].

While previous prospective studies involved patients who had shown a favorable response without disease progression, this study included some patients with oligo-progressive disease at the primary lung lesion according to our favorable clinical experience. In addition, this study showed promising results that are concordant with the results of previous studies. The 3-year OS and DFS rates of patients treated with consolidative SABR were 86.7% and 53.8%, respectively. During follow-up, in-field local recurrence occurred in only one patient (1/24, 4.16%) who showed stable disease in their tumor response evaluation. Interestingly, there was no local failure among patients who showed oligo-progressive disease at the thoracic site in their tumor response evaluation after systemic treatment.

With respect to treatment-related complications, most of the patients enrolled in this study had underlying lung diseases, such as COPD or IPF with impaired pulmonary function. Grade 3 radiation pneumonitis requiring hospitalization occurred in two patients (2/88, 2.3%), and there were no grade 4 or 5 treatment-related complications. Including one patient who showed acute exacerbation of COPD immediately after SABR, all patients were taking medication prior to SABR and were closely followed up with appropriate conservative treatment.

This study had some limitations. First, it was a retrospective analysis, and there may have been selection bias. Second, the sample size was too small to show a statistically significant difference between the two groups. Nonetheless, we evaluated the effect of SABR on the local control rates and the survival outcomes in patients with all stages of NSCLC according to the treatment aim.

## 5. Conclusions

Definitive SABR is appropriate for medically inoperable or high surgical risk patients with early stage NSCLC with acceptable treatment-related toxicities. Consolidative SABR improves local control rates and helps achieve long-term survival in patients with limited metastatic NSCLC.

## Figures and Tables

**Figure 1 medicina-58-01304-f001:**
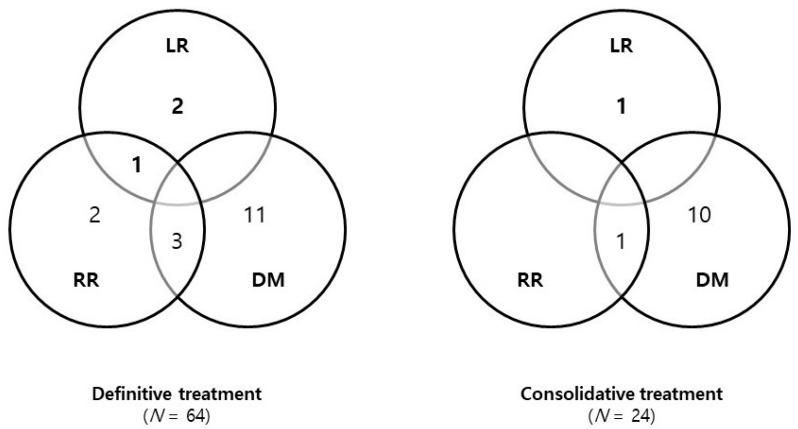
**Patterns of failure in patients with non-small cell lung cancer according to treatment aim.** Abbreviations: LR, local recurrence; RR, regional recurrence; DM, distant metastasis.

**Table 1 medicina-58-01304-t001:** Clinical characteristics according to treatment aim (*N* = 88).

Characteristics	Definitive (*n* = 64)	Consolidative (*n* = 24)
Age (years)	80 (61–91)	73 (53–89)
Sex		
Female	26 (40.6%)	8 (33.3%)
Male	38 (59.4%)	16 (66.7%)
Smoking Status		
Never smoker	32 (50.0%)	15 (62.5%)
Current or Ex-smoker	32 (50.0%)	9 (37.5%)
ECOG performance status		
0–1	42 (65.6%)	17 (70.8%)
2–3	22 (34.4%)	7 (29.2%)
Histology		
Adenocarcinoma	41 (64.1%)	16 (66.7%)
Squamous cell carcinoma	22 (34.4%)	7 (29.2%)
Others	1 (1.5%)	1 (4.1%)
Underlying lung disease		
COPD	22 (34.4%)	7 (29.2%)
IPF	1 (1.5%)	4 (16.7%)
None	41 (64.1%)	13 (54.1%)
Number of metastatic sites		
1		17 (70.8%)
2–3		7 (29.2%)
Prior treatment		
Platinum-doublet chemotherapy		13 (54.1%)
Target agents		11 (45.9%)
Immuno-oncologic agent treatment		
No		18 (75.0%)
Yes		6 (25.0%)
Response to systemic treatment		
Partial response		5 (20.8%)
Stable disease		15 (62.5%)
Oligo-progressive disease		4 (16.7%)

Data are presented as number (%) or median (range). Abbreviations: ECOG, Eastern Cooperative Oncology Group; COPD, chronic obstructive pulmonary disease; IPF, idiopathic pulmonary fibrosis.

**Table 2 medicina-58-01304-t002:** Clinical courses of patients who showed local recurrence after SABR.

No.	Age/Sex	Smoking	Histology	Treatment Aim	Underlying Disease	Time to LR (Months)	Follow-Up Period (Months)	Current Status
1	77/F	Never-smoker	ADC	Definitive	None	5, LR only	23	Alive with disease
2	78/M	Ex-smoker	SQCC	Definitive	COPD	15, LR + RR	35	
3	80/M	Ex-smoker	SQCC	Definitive	COPD	22, LR only	35	
4	79/M	Ex-smoker	SQCC	Consolidative	IPF	4, LR only	56	

Abbreviations: SABR, stereotactic ablative radiotherapy; ADC, adenocarcinoma; SQCC, squamous cell carcinoma; COPD, chronic obstructive pulmonary disease; IPF, idiopathic pulmonary fibrosis; LR, local recurrence; RR, regional recurrence.

**Table 3 medicina-58-01304-t003:** Clinical courses of patients requiring hospitalization after SABR.

No.	Age/Sex	PS	Smoking	FEV_1_/DLCO (pre-SABR)	Underlying Disease	Stage	FEV_1_/DLCO (Post-SABR)	Time to TRC (Months)
1	75/M	2	Ex-, 60 PY	1.5 L, 62%/56%	COPD, GOLD III	T1b N0	1.3 L, 47%/42%	3, RP
2	68/M	1	Ex-, 25 PY	2.6 L, 90%/24%	IPF	T2a N0	2.7 L, 92%/25%	3, RP/IPF AE
3	84/M	2	Ex-, 60 PY	0.8 L, 34%/40%	COPD, GOLD III	T2a N0	0.7 L, 31%/30%	1, COPD AE

Abbreviations: SABR, stereotactic ablative radiotherapy; PS, performance status; PY, pack-year; FEV1, forced expiratory volume in 1 s; DLCO, diffusing capacity of the lung for carbon monoxide; COPD, chronic obstructive pulmonary disease; GOLD, Global Initiative for Obstructive Lung Disease; IPF, idiopathic pulmonary fibrosis; TRC, treatment-related complications; RP, radiation pneumonitis; AE, acute exacerbation.

## Data Availability

The datasets used and/or analyzed during the current study are available from the corresponding author upon reasonable request.

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
