# Peer review of "Clinical Outcomes of Stereotactic Ablative Radiotherapy for All Stages of Non-Small Cell Lung Cancer; Definitive versus Consolidative"

_medicina, 2022, doi:10.3390/medicina58091304_

Round 1
Reviewer 1 Report
Dear authors,
thank you for the opportunity to review the manuscript "Clinical Outcomes of stereotactic ablative Radiotherapy for all stages of Non-Small Cell Lung Cancer; Definitive versus Consolidative” for Medicina.
You spend a lot of effort in preparation of this manuscript and are according to the STROBE checklist.
You cover an important topic and the results are very interesting.
I got few annotations:
-I think the comparism of two groups of different stages of NSCLC is not useful. Just like the p-valvue in the Kaplan Meier Curves and the annotations in the text. The groups should be stated seperate.
-The results of the authors are very interesting and the concept of additional SBRT after immuntherapy in patients with stage IV NSCLC is conclusive.
-The number of patients in the „consolidative“ group is low and further research is necessary.
-The number of patients with distant metastasis seem to be high. Only in the patients in stage IV these fits to the metastatic disease. Any explanations for the number of metastases in the „definitve“ group?
-The results in OS and DFS support the concept of additional SBRT after immuntherapy.
-Did the patients in stage I receive any chemo- or immuntherapy after SBRT?
Author Response
Please see the attachment.
Authors would appreciate the reviewer’s thoughtful comment and we prepared for the point-by-point response sincerely to your comments.

Reviewer 2 Report
Please refer to the attached file.

Author Response

(The authors gave the same response as above.)

Round 2
Reviewer 2 Report
Generally, the manuscript has been revised properly according to the reviewers’ comments. But some points are suggested to be modified before final acceptance.
- Table 1. Percentages of underlying lung disease in definitive group are still missing.
- Table 3. post-SBRT → post-SABR
- Abbreviations should be defined the first time they appear in each of three sections: the abstract; the main text; the first figure or table. When defined for the first time, the abbreviation should be added in parentheses after the written-out form.
→ Abbreviations of Figure 1 were not defined, although they appeared first time among Figures or Tables.
Author Response
Reviewer #2
Comments to the Author
General comments: Generally, the manuscript has been revised properly according to the reviewers’ comments. But some points are suggested to be modified before final acceptance.
→ Authors would appreciate the reviewer’s thoughtful comment and would like to apologize for not being able to thoroughly correct the error. We prepared for the point-by-point response sincerely to your comments.
Suggestions:
- Table 1. Percentages of underlying lung disease in definitive group are still missing.
→ Authors would like to apologize again for not being able to thoroughly correct the error and changes have been made according to the comment.
Revised manuscript, Table 1.
- Table 3. post-SBRT → post-SABR
→ Changes have been made according to the comment.
Revised manuscript, Table 3.
- Abbreviations of Figure 1 were not defined, although they appeared first time among Figures or Tables.
→ Authors would appreciate on this thoughtful comment. We added this information in Figure 1, as follows;
Abbreviations: LR, local recurrence; RR, regional recurrence; DM, distant metastasis.
Authors would appreciate the reviewer on this thoughtful comment again.
